# Imported malaria in Cabo Verde (2010–2024): Risks to post-elimination stability

**Adilson José DePina**[1]*, **Giovanni Leite Lima**[2], **António Lima Moreira**[3],
**El Hadji Amadou Niang**[4], **Klauss Kleydmann S. Garcia**[5]

**1** Programa Eliminação do Paludismo, CCS-SIDA, Ministério da Saúde, Praia, Cabo Verde,
**2** Universidade de Cabo Verde, Praia, Cabo Verde, **3** Programa Nacional de Luta contra as doenças de transmissão Vectorial e Problemas Ambientais, Ministério da Saúde, Praia, Cabo Verde, **4** Laboratoire d'Ecologie Vectorielle et Parasitaire (LEVP), Université Cheikh Anta Diop de Dakar, Dakar, Senegal, **5** University of Brasilia, Federal, Brazil

* Adilson.Pina@ccssida.gov.cv

## Abstract

Cabo Verde was officially certified malaria-free by the World Health Organization (WHO), following sustained public health interventions and strategic malaria elimination efforts. The country's National Strategic Plan (2020–2024) emphasized strengthening epidemiological and entomological surveillance at the archipelago's entry points (e.g., ports and airports), alongside early diagnosis and case investigation to prevent local transmission. However, imported malaria cases remain a persistent threat and challenge to prevent malaria reintroduction to maintain the elimination status. Therefore, this study aims to analyze imported malaria cases in Cabo Verde from 2010 to 2024, characterizing their locations, origins, epidemiological trends and spatial distribution. The findings aim to support evidence-based decision-making to prevent post-certification disease reintroduction. This study used an ecological time-series approach analyzing all confirmed imported malaria cases reported between 2010 and 2024 in Cabo Verde. Data was collected in collaboration with the National Malaria Control Program and the Integrated Surveillance and Response Service. Incidence, mortality, and case fatality rates were calculated. Joinpoint Regression Analysis was performed to assess time trends, and Holt-Winters additive models were applied for time-series forecasting. Spatial data visualization was also conducted. A total of 383 imported malaria cases were reported. A significant increase trend was observed from 2020 to 2024 (annual percentage change - APC): +25.75%). Forecast models estimate approximately 80 imported cases annually in 2025 and 2026 (-23.1 to 80). Most cases were reported in Santiago Island (68.9%), particularly in Praia (80.5%). The most common sources of imported infection were Guinea-Bissau (N = 90; 23,5%), Angola (N = 73; 19.1%), Senegal (49; 15.4%, and Nigeria (29; 7.6%). Malaria reintroduction risks persist in Cabo Verde, necessitating continuous surveillance and prevention efforts. Understanding the epidemiological trends and future

**Data availability statement:** The submission contains all the raw data required to replicate the study results. If required, additional details can be accessed by contacting the authors or by visiting https://minsaude.gov.cv/documentos/#all_0-114-relatorio-estatistico

**Funding:** The authors received no specific funding for this work.

**Competing interests:** The authors have declared that no competing interests exist.

projections is essential for maintaining Cabo Verde's malaria-free status. Vigilance and strategic interventions, including fast and correct case identification, treatment, and follow-up of imported cases, are some actions that need to be reinforced. Accurate policies, leadership capacity, and necessary resources are key requirements for maintaining and sustaining the elimination.

## Introduction

Cabo Verde is a member of Initiative E-2025 [1], whose goal is based on the World Health Organization's (WHO) Global Technical Strategy for Malaria 2016–2030 (GTS), which envisages the elimination of malaria in at least 25 countries by 2025 [2]. After six consecutive years without any local cases, Cabo Verde was officially certified by the WHO as a malaria-free country on January 12th, 2024. The WHO's certification of Cabo Verde as malaria-free is a testament to the power of strategic public health planning, collaboration, and sustained effort to protect and promote health [3]. This goal also recognises the latest success of the global fight against malaria. It is hoped that with existing and future tools, including vaccines, a malaria-free world will be achievable within this lifetime.

Certification of malaria elimination is the official recognition by WHO of a country's malaria-free status. It is granted when a country has shown, with rigorous, credible evidence, that the chain of Indigenous malaria transmission by *Anopheles* mosquitoes has been interrupted nationwide for at least the past three consecutive years. A country must also demonstrate the capacity to prevent the re-establishment of transmission [4].

The status of Cabo Verde as a malaria-free country is a result of the government's unwavering commitment and concerted efforts over the years. The successful implementation of the 2020/2024 National Strategic Plan for malaria elimination [5], which included strengthening epidemiological and entomological surveillance at ports and airports, is a testament to their dedication. The development of a comprehensive prevention plan [6] and the execution of various activities focused on early diagnosis and active case investigation further highlight their proactive approach.

Since discovering the islands, Cabo Verde has had a long history of malaria cases. There were times when the country interrupted local transmission of malaria cases, initially for five consecutive years in the 70s and the 80s, and for three successive years. Immediately after the interruptions, several malaria outbreaks jeopardized the gains made so far with the resurgence of autochthonous malaria cases [7]. This demonstrates, on the one hand, the inefficiency of the strategies implemented at the time to prevent reintroduction and the contribution of imported malaria cases to the failure to maintain a malaria-free status [3].

The World Health Organization (WHO) defines imported malaria as a "malaria case or infection that was acquired outside the area where it is diagnosed," provided the diagnosis is made within three months of returning from an endemic area [8]. Imported malaria consistently poses challenges to global elimination efforts. Studies

demonstrate that the movement of infected individuals, associated with other environmental conditions and vector capacities in receptive areas, has led to disease reintroduction post-elimination in some contexts [9,10].

Dealing with imported malaria poses significant challenges, and the ability of country-specific surveillance systems to diagnose and gather necessary data to determine importation status varies [11–13]. Many countries face challenges distinguishing between locally acquired and imported cases, which hinder cross-country collaboration in data collection, a crucial aspect in the fight against imported malaria [14–16]. Additionally, low-resource settings struggle with the impracticality of implementing molecular techniques required for tracking the introduction of drug-resistant parasites, potentially affecting the deployment of effective treatment. Inadequate surveillance is further complicated by the rapid mobility of infected individuals, which can change quickly in response to economic, political, climate, or other stressors. A country's ability to diagnose, identify, and ultimately treat imported malaria cases depends on robust surveillance and treatment systems capable of handling significant and sudden increases in malaria cases [9,15,16].

Identifying imported malaria cases varies widely among different countries, with each nation adopting unique approaches based on their surveillance capabilities, local malaria epidemiology, and geographical position relative to other malaria-endemic countries [17–19]. Additionally, the data collection and surveillance systems used to track imported malaria cases differ significantly from country to country, with some nations possessing advanced capabilities to identify and monitor cases while neighboring countries may not. Malaria surveillance data can be collected using a variety of approaches, including case-based surveillance, entomological surveys, and genetic surveillance. These approaches can help identify transmission patterns, assess the effectiveness of control measures, and prioritize interventions. It's important to note that the framework utilized to identify imported malaria cases can significantly impact a country's response to this public health issue [20,21].

In Cabo Verde, malaria classification follows national guidelines aligned with WHO directives [8]. For decades, especially with the pre-elimination Plan (2009–2013), all cases registered in the country have been the subject of an epidemiological investigation to better understand the case's history and apply the classification criteria [22].

Therefore, this study aims to analyze the trend of imported malaria cases into Cabo Verde from 2010 to 2024 to better understand the epidemiological scenario and describe its influence on the risk of malaria reintroduction. This will support decision-makers in selecting the best strategies to manage imported malaria cases and prevent their reintroduction in the country.

## Methodology

### Study design, population and period

This study undertakes an ecological time-series approach to analyze the trends and characteristics of imported malaria cases in Cabo Verde between 2010 and 2024. All confirmed imported malaria cases reported in the country during this period were quality-checked and then included in the analysis. According to the most recent demographic data, Cabo Verde is the third least populous country on the African continent. It has an estimated population of 511,534 inhabitants [23], with the majority residing in urban areas such as Santiago Island and São Vicente Island. The most significant proportion of Cape Verdeans live in the western region of the islands, which has better climatic and relief conditions, namely, temperate and less mountainous conditions. The island of Santiago, where the capital is located, is the most populous area of the archipelago and has approximately 270,000 inhabitants, more than 47% of the country's population. As a highly urbanized territory, 68% of Cape Verde's residents live in cities like Praia, the most populous town of the archipelago.

The population of Cabo Verde grows at a rate of 1.1% per year, with a birth rate three times higher than the death rate and a high fertility rate. The Cape Verdean population is young, with a median age of 28.8. Due to the improved implementation within strategic sectors such as health, the country's life expectancy is among the highest in Africa (74 years) [23].

For the purpose of this study, an imported malaria case was defined as "an infection acquired outside the country where it was diagnosed (Cabo Verde) [8]. The case definition includes a requirement for travel within a defined timeframe before the diagnosis.

## Study area

Cabo Verde is an archipelagic nation with ten islands and several islets off Africa's west coast. Its tropical climate has seasonal variations that influence the epidemiology of vector-borne diseases such as malaria. Due to its latitude and maritime conditions, Cabo Verde's climate varies from a temperate to the tropical bioclimatic zone. Average temperatures in the country are around 25 °C on the coast and 19 °C inland, especially in the highlands. In the warmer months, thermometers reach up to 30 °C, while cold periods register a minimum of 17 °C. The rainiest period of the year is from July to October, with monthly rainfall of approximately 50 millimeters. Rain in the country occurs due to the influence of hot, humid winds that blow from the northeast, from the European continent. Outside the rainy season, precipitation is very low and varies between three and seven millimeters per month [24].

## Data sources and collection

Malaria data reported in Cabo Verde from January 2010 to December 2024 were collected in collaboration with the National Malaria Control Program (NMCP) and the Integrated Surveillance and Response Service. These organizations compile national-level malaria data at the central level, which are cross-referenced with the statistical reports published by the Cabo Verde Ministry of Health over the years.

The variables included in the data encompassed case classification, demographic information (such as age, gender, and occupation), country of origin of malaria cases, *Plasmodium* species, and diagnosis locations at the municipality and island levels.

It is important to note that all malaria cases in Cabo Verde are diagnosed using rapid diagnostic tests and confirmed by microscopy. Furthermore, from 2018 onwards, the Medical Entomology Laboratory at the National Institute of Public Health has systematically confirmed all cases using polymerase chain reaction (PCR) testing.

The definitions of malaria cases and some classifications used in Cabo Verde are based on WHO definitions [8]. Table 1 provides definitions of key terms used in this study.

**Table 1. Malaria terminology used in malaria case classification in Cabo Verde, according to the WHO guidelines.**

| **Case definition** | |
| --- | --- |
| Malaria case | Occurrence of malaria infection in a person in whom the presence of malaria parasites in the blood has been confirmed by a diagnostic test. |
| Malaria case, confirmed | Malaria case (or infection) in which the parasite has been detected in a diagnostic test, i.e., microscopy, a rapid diagnostic test or a molecular diagnostic test. |
| **Case classification** | |
| Imported case | Malaria cases or infections in which the infection was acquired in a country different from Cabo Verde |
| Indigenous/local case | A case contracted locally (in Cabo Verde) with no evidence of importation and no direct link to transmission from an imported case. |
| Introduced case | A case contracted locally, with strong epidemiological evidence linking it directly to a known imported case (first-generation local transmission). |
| Recrudescent case | Recurrence of asexual parasitaemia of the same genotype(s) that caused the original illness, due to incomplete clearance of asexual parasites after antimalarial treatment. **Note**: *Recrudescence differs from reinfection with a parasite of the same or different genotype(s) and relapse in P. vivax and P. ovale infections.* |

## Data processing and statistical analysis

According to the guidelines, in Cabo Verde, all diagnosed malaria cases in health facilities are reported to the Health Delegation, which then informs the central level, namely the NMCP and the Surveillance Service. In turn, the NMCP and Surveillance Service conduct the necessary analyses, compile the data, prepare the required reports, and share them with other partners. Upon confirmation of a case, the Health Delegation team conducts a reactive investigation that includes home visits, rapid diagnostic tests in nearby houses, focal spraying, community sensitisation, and entomological data collection, allowing the classification of cases as local, imported, introduced or others, according to Table 1. The data was recorded and then processed in Microsoft Excel. Incidence rates were calculated as the total number of cases per 1,000 inhabitants, while mortality rates were computed as the total number of deaths per 1,000 inhabitants. The case fatality rates were determined as the proportion of deaths among confirmed cases, expressed as a percentage (total deaths/total cases × 100). All the analyses used the recent data from the national census [23].

## Trend analysis

Trends of the imported malaria cases were assessed using Joinpoint Regression Analysis, conducted in Joinpoint Regression Software (Version 4.9.1.0) [25]. The analysis employed Poisson distribution variance with heteroscedastic errors to detect the time series's significant inflection points (joinpoints). Annual Percent Changes (APC) were calculated and classified based on statistical significance ($p < 0.05$) as increasing, decreasing, or stable trends [26].

## Time-series forecasting

Future trends of putative imported malaria cases were estimated using the Holt-Winters [27] additive model, which is suitable for time-series that include zero or low values. Unlike the multiplicative version, the additive model does not assume proportional seasonal effects and therefore handles small or null monthly case counts without distortion. A seasonal period of 12 months was specified, reflecting the annual cyclic pattern observed in the data, with higher case counts consistently occurring in the second half of each year (S1 Table, S1 Fig).

To assess the adequacy of the time-series model, the Augmented Dickey-Fuller test was applied to test for stationarity. The result (Dickey-Fuller = -5.97, $p < 0.01$) indicates that the series is stationary, a necessary condition for reliable forecasting. Additionally, the Shapiro-Wilk normality test was used to evaluate whether the model residuals followed a normal distribution. The result (W = 0.989, p = 0.24) confirmed no significant deviation from normality, supporting the assumption of normally distributed errors (S2 Fig).

Forecasts were generated on a monthly basis for a 24-month horizon, including 95% confidence intervals (S3 Fig). All analyses were conducted using R language (version 4.1.1) and the forecast package [28].

## Spatial analysis and flow mapping

A spatial descriptive analysis was conducted using Quantum GIS (QGIS, Versions 2.18 and 3.34.3). Flow maps, generated using the Flowmaps Plugin (from QGIS version 2.18), visualized the movement of imported malaria cases from their country of infection (origin) to Cabo Verde (destination). The spatial descriptive analysis integrated a case matrix linking infected individuals to their geographic origins. Shapefile layers defining administrative boundaries were incorporated as provided by the National Institute of Territory Management. A global shapefile from the Opendatasoft database was also utilized for geospatial visualization [29].

## Ethics consideration

This study used secondary malaria data that is publicly available in different reports on the Ministry of Health website and others (https://minsaude.gov.cv/documentos/#all_0-114-relatorio-estatistico), as well as data from the NMCP reports

elaborated annually and shared with all the partners. The data ensured no individual's identification. Therefore, ethical approval was not required.

## Results

Data from the 15 years of study (2010–2024) reported 990 malaria cases nationwide. Most of the total cases registered were local (560; 56.6%), compared with imported (383; 38.7%). This is mainly due to the 2017 epidemic, in which 446 cases were reported, of which 423 were local. Still, in 2017, 18 relapse cases were from the local cases (Table 2).

There has been an alternation regarding the origin of the cases. During most of the study years, the number of imported cases exceeded that of local cases, from 2010 to 2013 and 2015. However, the opposite trend was observed in 2014, 2016 and 2017. After the last outbreak in 2017, with the interruption of local cases and the adaptation of case reporting, data analysis and investigation tools, it was possible after 2018 to record the introduced cases more accurately, with one case yearly, except in 2020 and 16 cases in 2024. During the study period, there were 26 recrudescence cases. The majority (18/26) were reported during the 2017 outbreak, with sporadic recrudescence cases also reported in 2013, 2016, 2019 and 2020.

Most of the imported malaria cases in the country were simple cases (357;93.2%), and almost all were caused by *Plasmodium falciparum* (377; 98.4%). The incidence rate among the general population varies between 0.02 in 2020 and 0.08 in 2019. A total of 6 deaths were reported from the diagnosed imported cases, with a mortality rate of 0.62% in 2011 and the highest case fatality rate of 10.3% in the same year. The cases were mainly men (314; 82.0%) and aged >25 (358; 93.5%) (Table 3).

### Imported malaria cases in Cabo Verde

The country reported 383 imported malaria cases during the studied period. The country reported most cases in 2012, 2019 and 2023, with 35, 39 and 36, respectively. Over the remaining years, the number of reported cases was less than 30, the lowest in 2020, with 10 cases (Fig 1). From 2010 to 2018, a decreasing trend is observed (APC: -5.30%). A sharp increase occurred between 2018 and 2019 (APC: + 93.73%), followed by a marked decline from 2019 to 2020 (APC: -59.90%), followed by a rebound from 2020 to 2024 (APC: + 22.67%).

During the study period, most (68.4%) of the malaria cases were reported in the second semester of the year, with the peak recorded toward the end of the year, in October, November, and December, with 15.9%, 11.2%, and 14.1%, respectively. The remaining months had between 3.0% and 10.4% of annual reported cases. (Fig 2).

In Fig 3, the forecast of imported cases in 2025 and 2026 can be visualized. The forecast predicts a slight increase in the number of imported cases, as shown in the upper limit of the 95% confidence interval (grey area in Fig 3). The predictions currently estimate approximately 30 (95% CI: −23.1; 84.0) cases for 2025 and 34 (95% CI: −28.0; 96.4) for 2026.

Throughout all years, most cases were reported in Santiago Island (261; 68.1%). Praia, the capital, accounted for 50.4% (209/383) of imported cases nationwide and 80.0% (206/261) of those within Santiago Island (Fig 4). São Vicente

**Table 2. Number of malaria cases reported in Cabo Verde during the study period, 2010–2024.**

| Characteristics | 2010 | 2011 | 2012 | 2013 | 2014 | 2015 | 2016 | 2017 | 2018 | 2019 | 2020 | 2021 | 2022 | 2023 | 2024 | TOTAL | % |
|---|---|---|---|---|---|---|---|---|---|---|---|---|---|---|---|---|---|
| Imported | 29 | 29 | 35 | 24 | 20 | 20 | 27 | 23 | 18 | 39 | 10 | 20 | 27 | 36 | 26 | 383 | 38,7 |
| Local | 18 | 7 | 1 | 23 | 24 | 7 | 45 | 423 | 1 | 0 | 0 | 0 | 0 | 0 | 11 | 560 | 56,6 |
| Introduced | 0 | 0 | 0 | 0 | 0 | 0 | 0 | 0 | 1 | 1 | 0 | 1 | 1 | 1 | 16 | 21 | 2,1 |
| Recrudescence | 0 | 0 | 0 | 3 | 0 | 0 | 3 | 18 | 0 | 1 | 1 | 0 | 0 | 0 | 0 | 26 | 2,6 |
| TOTAL | 47 | 36 | 36 | 50 | 44 | 27 | 75 | 464 | 20 | 41 | 11 | 21 | 28 | 37 | 53 | 990 | 100,0 |
| % | 4,7 | 3,6 | 3,6 | 5,1 | 4,4 | 2,7 | 7,6 | 46,9 | 2,0 | 4,1 | 1,1 | 2,1 | 2,8 | 3,7 | 5,4 | 100 | – |

**Table 3. Imported malaria cases in Cabo Verde between 2010 and 2024.**

| Character-istics | 2010 | 2011 | 2012 | 2013 | 2014 | 2015 | 2016 | 2017 | 2018 | 2019 | 2020 | 2021 | 2022 | 2023 | 2024 | TOTAL | % |
|---|---|---|---|---|---|---|---|---|---|---|---|---|---|---|---|---|---|
| Simples | 28 | 24 | 27 | 20 | 19 | 20 | 26 | 23 | 18 | 39 | 10 | 20 | 24 | 34 | 25 | 357 | 93,2 |
| Complicated | 1 | 5 | 8 | 4 | 1 | 0 | 1 | 0 | 0 | 0 | 0 | 0 | 3 | 2 | 1 | 26 | 6,8 |
| **Cases by Plasmodium species** | | | | | | | | | | | | | | | | | |
| Pl. falciparum | 29 | 29 | 35 | 24 | 20 | 20 | 27 | 23 | 18 | 38 | 8 | 20 | 27 | 33 | 26 | 377 | 98,4 |
| Pl. vivax | 0 | 0 | 0 | 0 | 0 | 0 | 0 | 0 | 0 | 0 | 0 | 0 | 0 | 0 | 0 | 0 | 0,0 |
| Pl. ovale | 0 | 0 | 0 | 0 | 0 | 0 | 0 | 0 | 0 | 0 | 2 | 0 | 0 | 0 | 0 | 2 | 0,5 |
| Mixed (Pf+Pm) | 0 | 0 | 0 | 0 | 0 | 0 | 0 | 0 | 0 | 1 | 0 | 0 | 0 | 3 | 0 | 4 | 1,0 |
| **Incidence, mortality and Case fatality rate** | | | | | | | | | | | | | | | | | |
| Incidence rate | 0,06 | 0,06 | 0,07 | 0,05 | 0,04 | 0,04 | 0,05 | 0,05 | 0,04 | 0,08 | 0,02 | 0,04 | 0,05 | 0,07 | 0,05 | 0 | – |
| Death | 0 | 3 | 1 | 0 | 0 | 0 | 0 | 0 | 0 | 0 | 0 | 0 | 1 | 1 | 0 | 6 | 10 |
| Mortality rate | 0,00 | 0,62 | 0,21 | 0,00 | 0,00 | 0,00 | 0,00 | 0,00 | 0,00 | 0,00 | 0,00 | 0,00 | 0,20 | 0,20 | 0,00 | 0 | – |
| Case fatality rate (%) | 0,0 | 10,3 | 2,9 | 0,0 | 0,0 | 0,0 | 0,0 | 0,0 | 0,0 | 0,0 | 0,0 | 0,0 | 3,7 | 2,8 | 0,0 | 0 | – |
| Population | 477 859 | 480 577 | 483 285 | 485 996 | 488 719 | 491 436 | 493 465 | 495 522 | 497 558 | 499 609 | 501 657 | 504 125 | 506 595 | 509 078 | 511 534 | | |
| **Cases by sex** | | | | | | | | | | | | | | | | | |
| Male | 22 | 24 | 29 | 21 | 17 | 17 | 21 | 18 | 17 | 28 | 9 | 17 | 24 | 29 | 21 | 314 | 82,0 |
| Female | 7 | 5 | 6 | 3 | 3 | 3 | 6 | 5 | 1 | 11 | 1 | 3 | 3 | 7 | 5 | 69 | 18,0 |
| **Cases by age group** | | | | | | | | | | | | | | | | | |
| 0 - 4 Years | 0 | 1 | 0 | 1 | 0 | 0 | 0 | 0 | 0 | 0 | 1 | 0 | 0 | 0 | 0 | 3 | 0,8 |
| 5 - 9 Years | 0 | 0 | 0 | 0 | 0 | 0 | 0 | 0 | 0 | 0 | 0 | 0 | 0 | 0 | 0 | 0 | 0,0 |
| 10 - 14 Years | 0 | 0 | 0 | 0 | 0 | 0 | 1 | 0 | 0 | 2 | 0 | 0 | 7 | 1 | 0 | 11 | 2,9 |
| 15 - 19 Years | 0 | 1 | 2 | 1 | 2 | 0 | 1 | 0 | 1 | 1 | 0 | 0 | 0 | 2 | 0 | 11 | 2,9 |
| 20 Years + | 29 | 27 | 33 | 22 | 18 | 20 | 25 | 23 | 17 | 36 | 9 | 20 | 20 | 33 | 26 | 358 | 93,5 |
| **TOTAL** | **29** | **29** | **35** | **24** | **20** | **20** | **27** | **23** | **18** | **39** | **10** | **20** | **27** | **36** | **26** | **383** | **100** |

came in the second position regarding the number of imported cases (12.6%), followed by Sal (8.4%). Santa Catarina in Santiago and São Filipe in Fogo reported 6.4% and 3.9%, respectively, during the same period.

The analysis of the origin of the imported malaria cases in Cabo Verde highlighted that Guinea-Bissau (N = 90; 23,4%), Angola (N = 73; 19.1%), Senegal (N = 59; 15.4%) and Nigeria (N = 29; 7.6%) are the countries that contribute the most to the importation of malaria into the archipelago. Other countries contributed at a lower level to the imported cases into the country, as shown below (Figs 5, 6 and in S1 Video).

## Discussion

The findings of this study indicate notable shifts in the pattern of imported malaria cases in Cabo Verde over the past 15 years. Between 2010 and 2018, a consistent decline was observed, followed by a sustained increase in cases from 2020 onward. Forecasting analyses further suggest that this upward trend may continue in the near future, underscoring the need for timely and effective diagnostic strategies to prevent onward transmission and safeguard the country's malaria-free status. Moreover, the data reveal that most imported cases originate from a limited number of countries

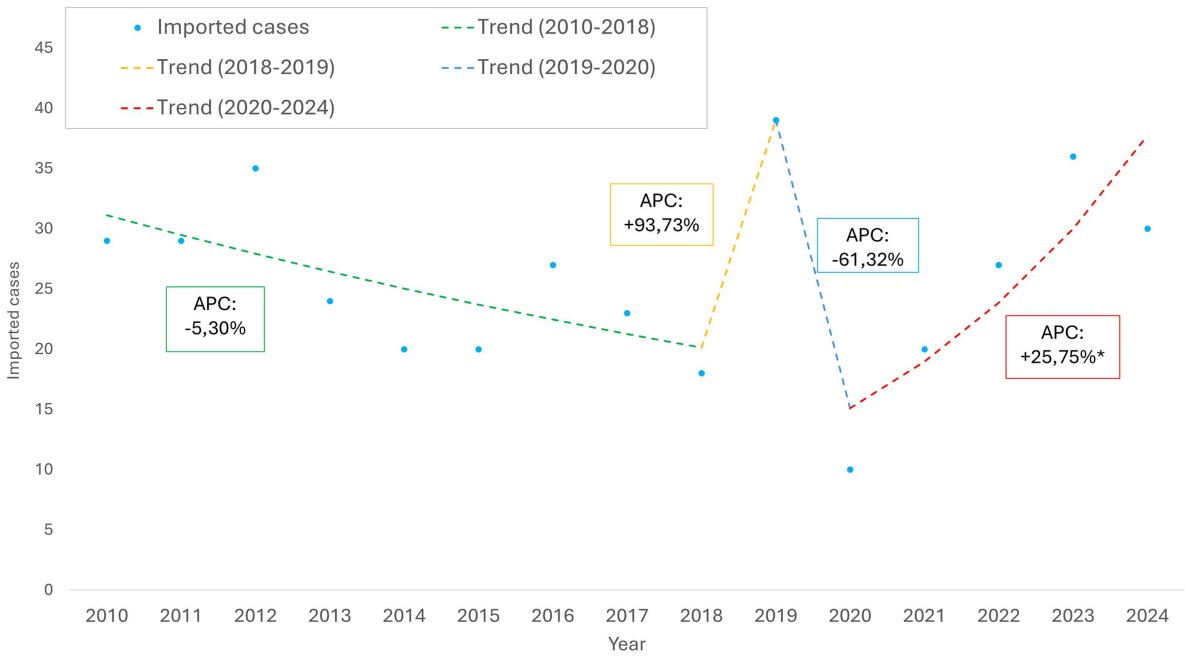

**Fig 1. Imported malaria cases and temporal trends in Cabo Verde by years, 2010–2024.**

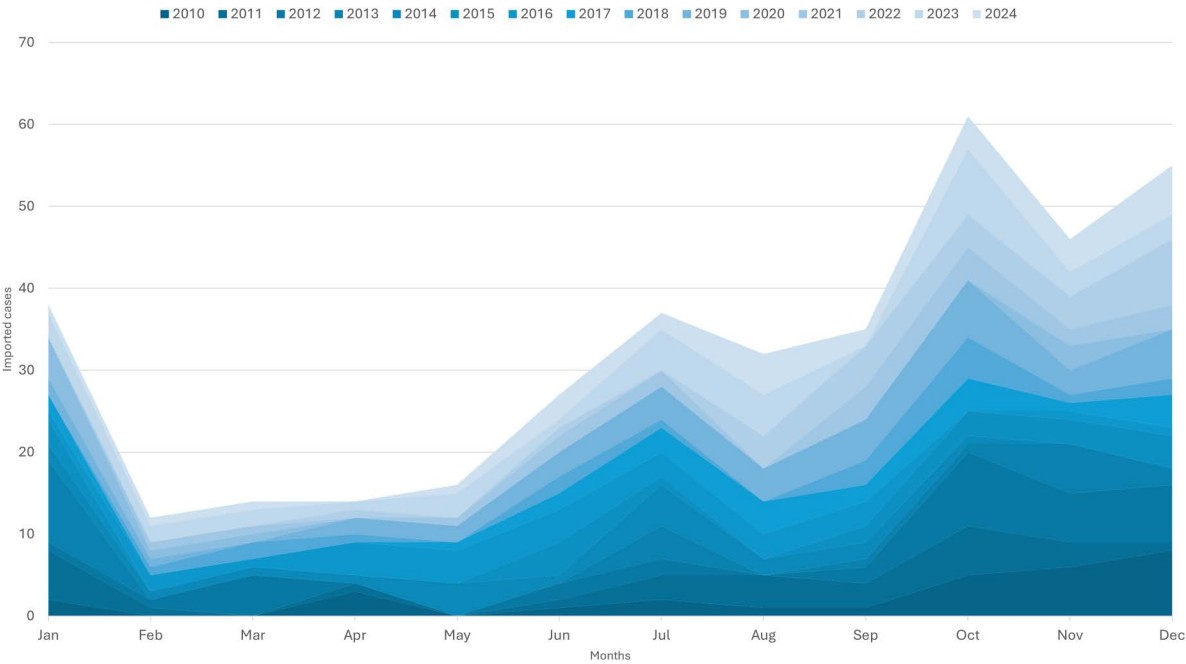

**Fig 2. Stacked chart of imported malaria cases seasonality in Cabo Verde by month, 2010–2024.**

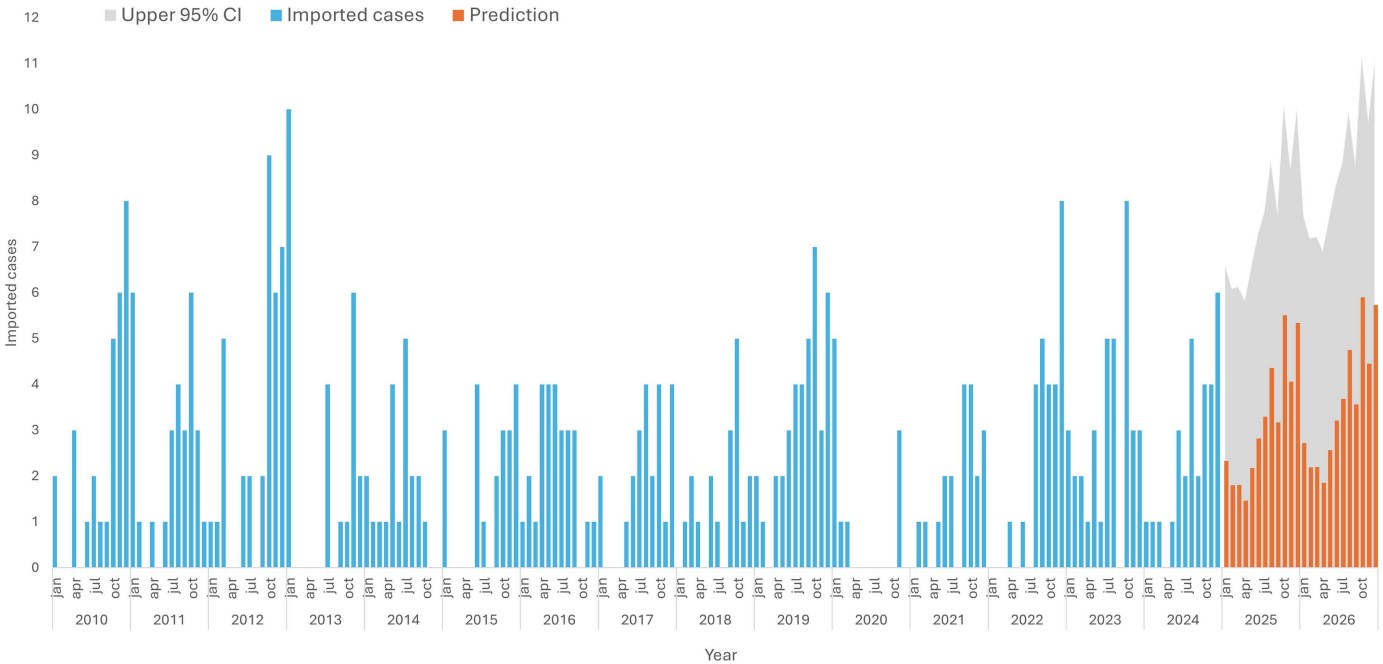

**Fig 3. Temporal prediction of imported malaria cases in Cabo Verde for 2025 and 2026.**

- particularly Guinea-Bissau, Angola, and Senegal. This information is critical for informing cross-border collaboration. It could support Cabo Verde's Ministry of Health in engaging with health authorities in these countries to develop targeted travel medicine strategies to sustain elimination.

Using joinpoint regression methods to analyse the time-series behaviour provided an appropriate understanding of the temporal changes in imported malaria trends in Cabo Verde. Unlike descriptive analyses, which can illustrate overall fluctuations, joinpoint regression formally identifies statistically significant changes in trend direction. It estimates the Annual Percentage Change (APC) within each time segment, highlighting that imported cases have risen in the country since 2020. The analysis revealed a marked increase in imported cases after 2019, with a +25.75% APC, highlighting a reversal in the declining trend. Similar methods should be conducted in other countries to understand malaria's time-series behaviour.

Certification of malaria elimination is the official recognition by WHO of a country's malaria-free status. It is granted when a country has shown, with rigorous, credible evidence, that the chain of indigenous malaria transmission by Anopheles mosquitoes has been interrupted nationwide for at least the past three consecutive years. A country must also demonstrate the capacity to prevent the re-establishment of transmission [8].

Cabo Verde achieved malaria elimination and was certified as a malaria-free country by the WHO in 2024. This achievement is a testament to the importance of political engagement, strategic public health planning, collaboration, and sustained effort to protect and promote health. Cabo Verde's success was built upon the following five critical pillars: i) a strong political engagement; ii) an adequate surveillance and rapid response system; iii) a multi-sectoral and community-based approach, where the government worked closely with the community; iv) international partnerships with organisations such as WHO and the Global Fund; and v) diligence – once the disease appears eradicated, a plan to prevent re-establishment is required [30]. Moreover, this achievement is the result of work carried out by the country over the years. To sustain the achieved malaria-free status, the country developed a 2020/2024 work plan to reinforce the epidemiological and entomological surveillance at the main entry point (ports and airports) of the archipelago´s as one of the

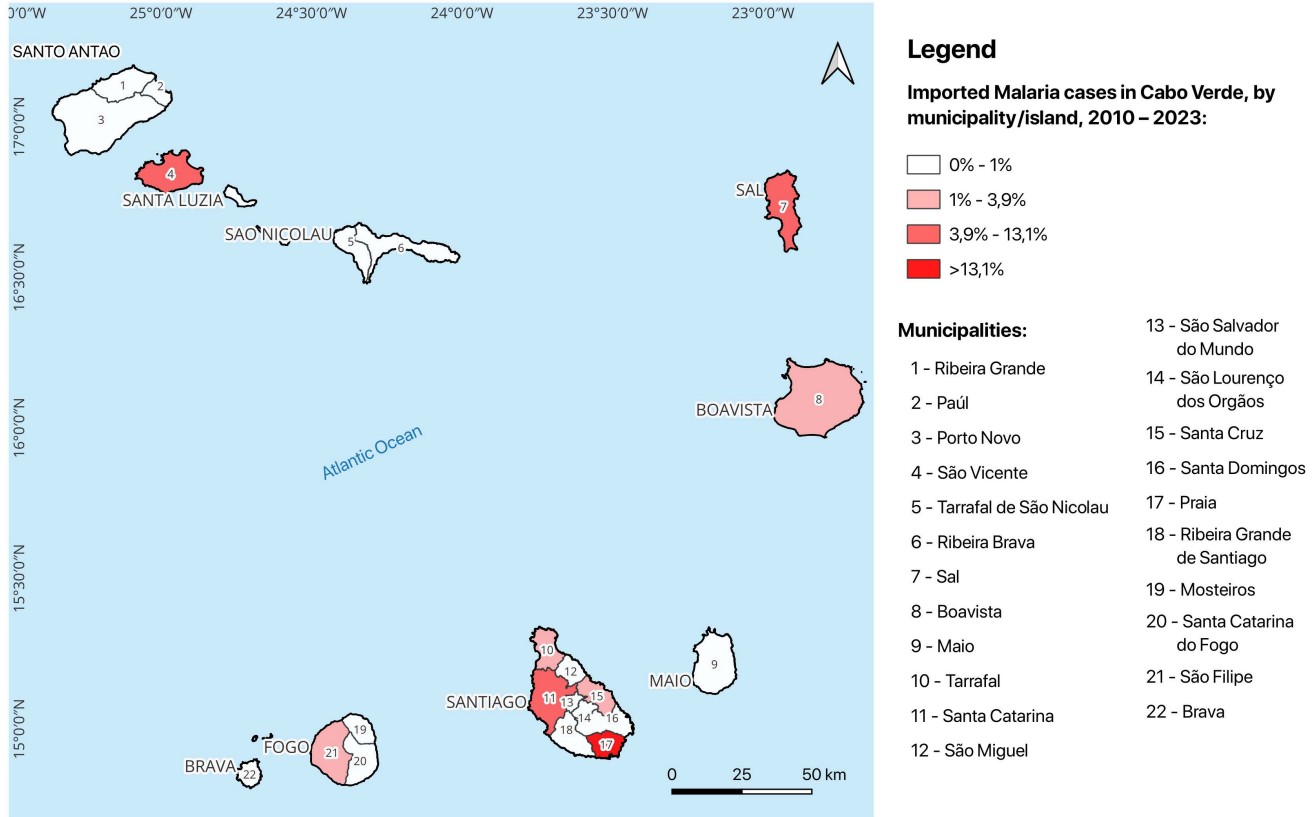

**Fig 4. Imported malaria cases by municipality/island, in Cabo Verde, 2010–2024.** Source: The map was created using the USGS National Map Viewer (https://www.usgs.gov/tools/national-map-viewer).

mandatory requirements to be certified. Cabo Verde commits to complying with a prevention plan, which includes several activities regarding early diagnosis and case investigation to avoid local cases.

Recent studies show that livelihood activities and population mobility influence the epidemiology of vector-borne disease, which may increase the risk of malaria cases, and are barriers to malaria in elimination settings [31,32]. Imported malaria cases pose a recognized threat to malaria-free settings, as they can serve as a source of reintroduction when competent vectors are present and local transmission conditions remain favorable. Studies have shown that even a single imported case, if undetected or poorly managed, can trigger localized outbreaks and compromise elimination gains, particularly in areas with persistent vector populations and climatic suitability for transmission [10,15]. This underscores the importance of maintaining robust border surveillance, rapid diagnostic capacity, and timely case investigation, especially in island contexts like Cabo Verde, where entry points are concentrated and mobility patterns can be mapped.

Regarding the imported cases, it is worth highlighting that Cabo Verde has always had a strong emigrant community in several African malaria-endemic countries, maintaining the link with the relatives they visit during holidays. This explains the observed imported malaria cases mainly from Angola and Guinea-Bissau, both African Portuguese-speaking countries and members of PALOP (African Countries of Portuguese Official Language) organization, which are the privileged destination of the Cape-Verdeans. In addition to the "brother countries," Cabo Verde is also an ECOWAS (Economic Community of West African States) member with open borders to neighboring countries like Senegal and Guinea-Bissau, among others. As a result, Cabo Verde has become a preferred destination due to the increasing mobility in the region. The increase in flights and mobility over the last few years has increased the risk of malaria importation into the archipelago.

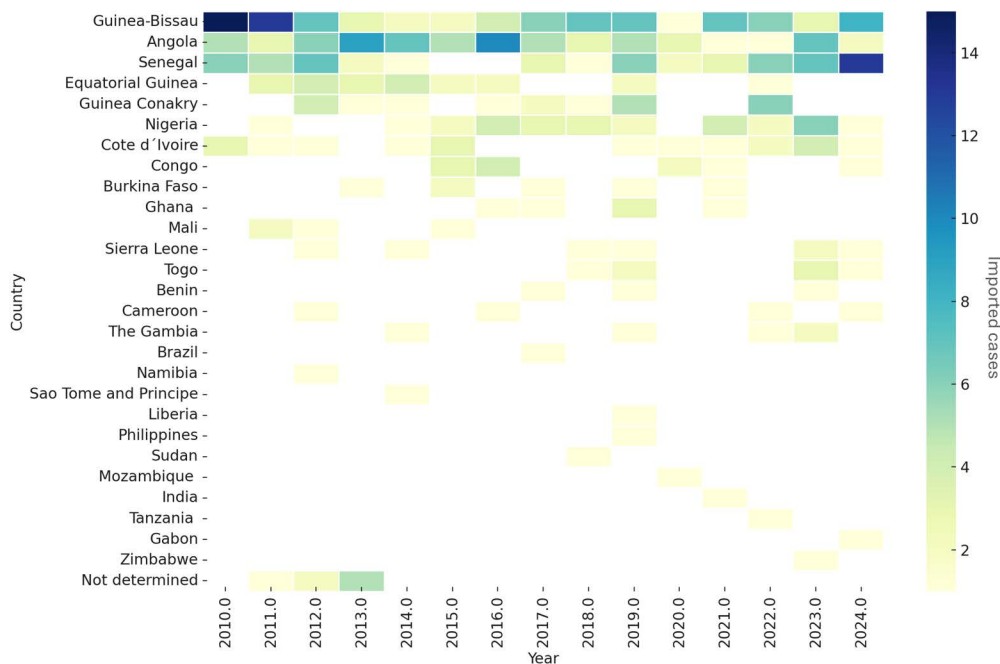

**Fig 5. Countries contributing the most to Cabo Verde imported malaria cases, 2010–2024.**

As mentioned in several other studies [33–38], the predominant species of imported malaria in several countries is *P. falciparum*, as reported here. Similarly to elsewhere on the African continent, most of the imported malaria cases originated from some of the most malaria-endemic African countries, where *P. falciparum* is the predominant parasitic species in humans [39,40].

As a country that has recently achieved malaria-free status, Cabo Verde must continue its efforts to prevent malaria re-introduction. This requires creating conditions to better understand the root causes of the recorded imported cases and developing tailored solutions to prevent malaria importation. One of the challenges in the country is the lack of a critical mass of qualified human resources at all levels of the health pyramid. It is vital to reinforce the workforce and their skills/competencies within the health system that should be integrated, have a holistic vision and keep the awareness that malaria remains a public health issue [41,42]. As an insular country with 22 municipalities, hundreds of health structures, and various levels of care, reinforcement of capacity in malaria case management becomes essential. Regarding the epidemiology of the disease over the recent years [43,44], we should distinguish the few municipalities, such as the islands of Santiago, São Vicente, Sal and Boavista, which frequently reported cases. On the other hand, several other islands like Santo Antão, São Nicolau, Maio and Brava, have not reported cases for decades. Hence, there is a need to strengthen skills at all levels to ensure that technicians are sufficiently prepared to undertake correct, rapid diagnosis for accurate treatment and thus prevent imported cases from initiating local transmission.

Tourism is a vital sector in Cabo Verde, accounting for approximately 25% of the country's Gross Domestic Product (GDP). In 2023, Cabo Verde welcomed over one million tourists, representing a 20.9% increase compared to the previous year. As a popular tourist destination, the risk of malaria for travellers does not always reflect the risk faced by the local population [13]. Therefore, effective surveillance of travel-related malaria cases is essential. Additionally, malaria is often associated with poverty [45], and recent studies have highlighted socioeconomic inequalities in the burden of the disease [45,46]. While achieving malaria elimination in Cabo Verde demonstrates progress in health, it also reflects advancements in various determinants of malaria, such as education, socioeconomic status, environmental factors and others.

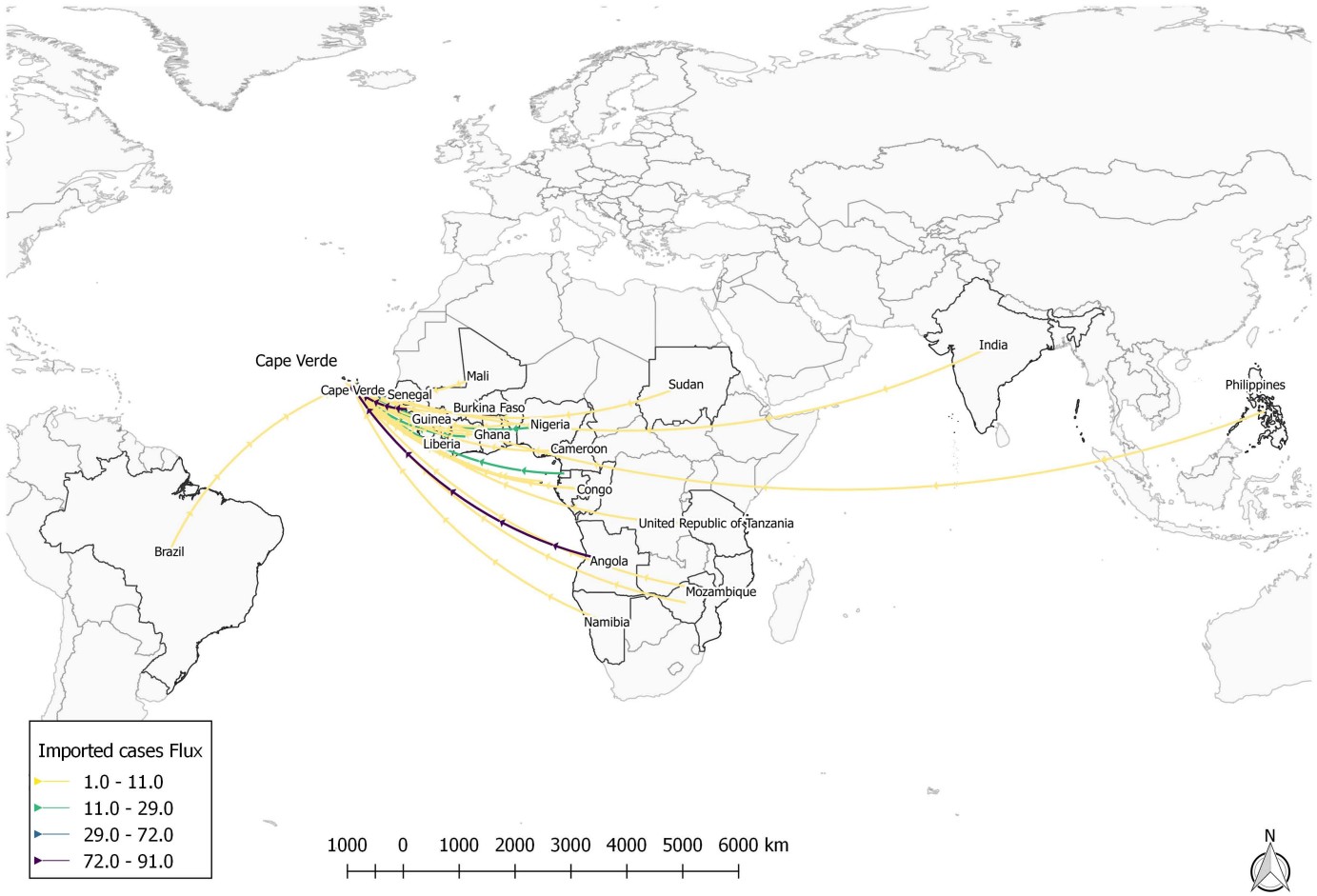

**Fig 6. Imported malaria cases in Cabo Verde by country of origin, 2010–2024.** Source: Opendatasoft. Shapefile available at: https://public.open-datasoft.com/explore/dataset/world-administrative-boundaries/export.

Another challenge resulting from the malaria certification related to the response against the imported cases is integrating service delivery for malaria response in the national health information service. In this sense, the country must prioritize integrated service delivery to advance malaria reintroduction prevention effectively. This requires a strong focus on strengthening health systems, ensuring the availability and use of high-quality data at all levels, and addressing equity issues [43]. Cabo Verde must promote operational research and innovation for new tools, deepen the understanding of effective implementation strategies for interventions, and make a compelling case for investing in malaria among stakeholders. Engaging with impacted communities nationwide is essential to drive these initiatives forward.

Malaria data in the country reveal important insights into patient care-seeking behaviour and the effectiveness of health interventions. On average, there is a three-day delay between the onset of malaria symptoms and when the patient visits the nearest health facility, with this interval ranging from zero to nine days. Indigenous cases tend to have a slightly shorter delay, averaging 2.6 days before patients seek care, whereas imported cases exhibit a longer delay of approximately 4.2 days [47]. In 2016, the situation was particularly pronounced in Praia, the most affected community, where the average time from symptom onset to health service visit was notably lower, at just 2.1 days. This shorter response time suggests that factors such as community awareness and accessibility of healthcare services may play a significant role in faster diagnosis and treatment [47].

The longer delay observed among imported cases carries significant implications for malaria control and elimination efforts. One likely explanation is that individuals infected abroad may not recognize malaria as a possible cause of their symptoms, particularly fever, due to a lack of prior exposure or limited knowledge about the disease. This unawareness can lead to delayed care-seeking, increasing the risk of clinical complications and, in elimination settings, heightening the potential for local transmission if cases go undetected. Travel medicine interventions could be strengthened at ports and airports to address this challenge. For example, screening symptomatic travelers or individuals arriving from endemic regions may help shorten the time to diagnosis and facilitate earlier treatment, ultimately reducing the risk of reintroduction [48,49].

Despite findings from the knowledge, attitudes, and practices (KAP) study indicating in the country, that a commendable 88% of the population intends to seek medical attention within 48 hours of first experiencing symptoms, there remains a marked discrepancy between this knowledge and actual practical care-seeking behavior [48]. This gap highlights the need for enhanced public health messaging and Behavioral Change Communication (BCC) interventions to ensure timely care-seeking behavior. Additionally, the origin of malaria cases is a significant factor regarding disease control strategies. Most imported cases were found among travellers from African malaria-endemic countries, besides notable contributions from Brazil and several Asian countries, complicating the public health response.

The presence of other malaria species, such as *P. vivax* and *P. ovale*, poses additional challenges, as these can result in more severe impacts on the population [50,51]. Therefore, effective malaria control strategies must address these complexities to mitigate the risks associated with different malaria strains and improve health outcomes for affected communities. Although the occurrence of *P. ovale* and mixed infections is low in the country, it is vital that health surveillance training be maintained to keep health professionals alert to rapidly and properly detect and diagnose these infections, to avoid inappropriate treatment, which could lead to severe forms or recurrences.

The imported malaria cases in Cabo Verde have a specific gender and age profile, mainly (82%) male, over 25. Although there is no specific characterization at the level of professions, data points mainly to trips related to business and professionals linked to long-distance travel from Cabo Verde to endemic countries on the continent. This data is similar to that of other studies [52,53] and demonstrates that understanding these demographics is crucial for designing effective malaria prevention and treatment strategies.

Notwithstanding the goal achieved, Cabo Verde's malaria-free status also shows that the fight against malaria is far from being over. The country should be aware of the impacts of climate change on malaria, taking into account realities around the world and cases appearing in places where the disease had previously been eradicated or had not appeared before [54]. Cabo Verde should reinforce its global strategies and adapt its response to mitigate the risks associated with climate change and malaria transmission. This implies improved Surveillance Systems and enhanced monitoring systems to track the above-mentioned changes in mosquito populations and/or malaria incidences and climatic conditions. Community Awareness Programs, through Educating the communities about preventive measures against mosquito bites, will be crucial as conditions change.

Moreover, with the 'climate change' risks, fast reintroduction can happen if any imported case fails to be detected and treated [55–57]. Malaria cases in the African continent may increase, which can influence an increase of imported cases into Cabo Verde – which also aligns with the trend analysis and the prediction models presented. So, now, more than ever, Cabo Verde should reinforce active imported cases surveillance and not 'relax' due to the low absolute number of cases. Staying alert will be the key to maintaining the country as a malaria-free territory.

Given these findings, the persistent risk of malaria reintroduction in Cabo Verde necessitates ongoing and robust public health interventions. Continuous surveillance and effective prevention strategies are crucial to mitigate the threat of imported malaria and safeguard the population's health. Strong prevention efforts drive success by providing vector control, chemoprevention, and robust, resilient health systems that ensure everyone can access quality malaria diagnosis and treatment services. Community health workers are critical in eliminating malaria, particularly in hard-to-reach areas.

Political commitment is also crucial to ensure that sufficient resources and effective leadership are sustained until the very end, and – beyond elimination – to prevent the re-establishment of malaria.

Some limitations are worth acknowledging. First, the analysis relies on secondary data from the national surveillance system managed by Cabo Verde's Ministry of Health. While this is the official and most reliable source of malaria case reporting in the country, routinely collected data may still be subject to underreporting, misclassification, or delays in notification. Second, the ecological design of the study limits the ability to conclude at the individual level regarding risk factors or behavioral determinants. However, this approach remains appropriate for assessing long-term trends and informing population-level public health strategies. Finally, the forecasting analysis is based on historical data and assumes that current patterns will remain stable. This assumption may not hold in the face of unexpected disruptions, such as major climatic events, political instability, or travel and migration patterns, which could influence future importation trends in either direction.

## Conclusions

Imported malaria cases have become a challenge in Cabo Verde throughout the different phases of fighting against the disease, from control to elimination. Despite the gains made over recent years, which contributed to the certification of Cabo Verde as a malaria-free country, imported malaria cases have been variable and pose a serious threat to the archipelago. There has been constant reporting of cases over the years, originating mainly from African malaria-endemic countries within the sub-region. Effective anticipation efforts are key to successfully preventing malaria reintroduction. These efforts include active surveillance and reactive control measures while establishing strong, resilient health systems that provide everyone with access to high-quality malaria diagnosis and treatment services. Community health workers play a vital role in sensibilization, and political commitment is essential to maintain adequate resources and strong leadership. In summary, implementing effective strategies adapted to the context of a malaria-free country is crucial to prevent malaria reintroduction in Cabo Verde.

## Supporting information

**S1 Table. Time series structure – Imported malaria cases in Cabo Verde.**
(XLSX)

**S1 Fig. Decomposition of Cabo Verde's imported malaria cases: Additive time-series model.**
(TIFF)

**S2 Fig. Residual Analysis of Cape Verde's imported malaria cases: Additive time-series model.**
(TIFF)

**S3 Fig. Cross-correlation plots among forecasted point values and their associated 95% prediction intervals.**
(TIFF)

**S1 Video. Time behavior of imported malaria cases into Cabo Verde, 2010–2024.**
(MP4)

## Acknowledgments

We thank all authors for their contributions, the National Malaria Control Program, and the Surveillance service for making the data available for analysis.

## Author contributions

**Conceptualization:** Adilson José DePina, Giovanni Leite Lima, El Hadji Amadou Niang.

**Data curation:** Giovanni Leite Lima, Klauss Kleydmann S. Garcia.

**Formal analysis:** Adilson José DePina, Giovanni Leite Lima, El Hadji Amadou Niang, Klauss Kleydmann S. Garcia.

**Funding acquisition:** Adilson José DePina, António Lima Moreira.

**Investigation:** Adilson José DePina, El Hadji Amadou Niang, Klauss Kleydmann S. Garcia.

**Methodology:** El Hadji Amadou Niang.

**Project administration:** Adilson José DePina, António Lima Moreira.

**Resources:** Adilson José DePina, António Lima Moreira.

**Software:** Klauss Kleydmann S. Garcia.

**Supervision:** Adilson José DePina, El Hadji Amadou Niang.

**Validation:** Adilson José DePina, Giovanni Leite Lima, António Lima Moreira, El Hadji Amadou Niang, Klauss Kleydmann S. Garcia.

**Visualization:** Adilson José DePina, Giovanni Leite Lima, El Hadji Amadou Niang, Klauss Kleydmann S. Garcia.

**Writing – original draft:** Adilson José DePina, El Hadji Amadou Niang, Klauss Kleydmann S. Garcia.

**Writing – review & editing:** Adilson José DePina, Giovanni Leite Lima, António Lima Moreira, El Hadji Amadou Niang, Klauss Kleydmann S. Garcia.

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
