## [Decision Letter · Decision Letter 0]

7 Apr 2025

PGPH-D-25-00516

Characterization of Imported malaria in Cabo Verde, 2010-2024: A challenge in preventing the disease re-establishment in the country

Dear Dr. DePina,

Thank you for submitting your manuscript to PLOS Global Public Health. After careful consideration, we feel that it has merit but does not fully meet PLOS Global Public Health’s publication criteria as it currently stands. Therefore, we invite you to submit a revised version of the manuscript that addresses the points raised during the review process.

The first area of focus for the revision should be the presentation clarity. This includes the definition of terminology which should be consistent with what is presented in the tables. Secondly, as mentioned by a reviewer, the methods (joinpoint regression and time series analysis) should be described in such a manner that it is reproducible. To that end, it would be good to detail the data processing, methods (including model selection and goodness-of-fit measures) and results.   

We look forward to receiving your revised manuscript.

Kind regards,

Michele Nguyen

Academic Editor

Journal Requirements:

1. We note that your Data Availability Statement is currently as follows: All data are available on the manuscript.

2. Figures 4 and 6: please (a) provide a direct link to the base layer of the map (i.e., the country or region border shape) and ensure this is also included in the figure legend; and (b) provide a link to the terms of use / license information for the base layer image or shapefile. We cannot publish proprietary or copyrighted maps (e.g. Google Maps, Mapquest) and the terms of use for your map base layer must be compatible with our CC-BY 4.0 license. 

Additional Editor Comments (if provided):

Reviewers' comments:

Reviewer's Responses to Questions

**Comments to the Author**

1. Does this manuscript meet PLOS Global Public Health’s publication criteria?

Reviewer #1: Yes

Reviewer #2: Yes

Reviewer #3: Yes

Reviewer #4: Partly

Reviewer #5: Partly

2. Has the statistical analysis been performed appropriately and rigorously?

Reviewer #1: Yes

Reviewer #2: Yes

Reviewer #3: Yes

Reviewer #4: I don't know

Reviewer #5: N/A

3. Have the authors made all data underlying the findings in their manuscript fully available (please refer to the Data Availability Statement at the start of the manuscript PDF file)?

Reviewer #1: Yes

Reviewer #2: Yes

Reviewer #3: Yes

Reviewer #4: No

Reviewer #5: Yes

4. Is the manuscript presented in an intelligible fashion and written in standard English?

Reviewer #1: Yes

Reviewer #2: Yes

Reviewer #3: Yes

Reviewer #4: Yes

Reviewer #5: Yes

Reviewer #1: Summary:

This paper examines malaria cases in Cabo Verde over a 14-year period leading up to 2024, when the country achieved WHO certification for malaria elimination. During this period, Cabo Verde operated as a near-elimination setting, primarily focusing on the challenge of imported cases. The analysis includes calculations of incidence, mortality, and case-fatality rates, along with trend analysis using joinpoint regression. Publishing these findings is valuable for other countries approaching elimination, as it may provide insights on achieving and maintaining this goal.

Despite a few minor discrepancies in the results section, I recommend this paper for publication.

MAJOR COMMENTS:

Results

Table 1 and First Paragraph:

In the text, you mention 441 local cases in 2017, but in Table 1, this is broken down into 423 local cases and 18 relapsed cases. It’s important to clarify the terminology used if relapsed are also local or not and update both the text and the table accordingly.

Second Paragraph:

The authors state that after 2017, the country began registering only imported cases, but this is not reflected in Table 1. There is one local case in 2018 and 11 in 2014, which is also mentioned in the following paragraph. Please revise or remove this statement. Additionally, the authors mention that after 2018, it became possible to more accurately record imported cases, with only one case per year. However, in Table 1 there are more than one per year. Please check the text and Table 1 to ensure they align and clearly define "introduced" and "relapsed" cases.

MINOR COMMENTS

Abstract

- Please spell out all acronyms and, where possible, avoid using them in the abstract.

- Angola (N = 72; 18.6%%) – additional percentage should be removed.

Introduction

Paragraph 4

- The second sentence is unclear. Please revise: “There were times when the country interrupted local cases transmission.”

- In the same sentence you mention there were times when the country interrupted local transmission could you provide any evidence in the form of citations for these.

- In the third sentence, please clarify or correct the phrase “Immediately after the outages.” What do you mean by "outages"?

Methods

Study design - Paragraph 2

- The total population is written as 511.534; I assume this should be 511,534.

- When describing where most people live, you mention these areas have better climatic and relief conditions. Please clarify why they are considered better—are they more temperate and less mountainous?

- When describing the country's demography, could you include references for age, population growth rate, and life expectancy?

- Please clarify which portion of 2024 the data covers. Is it for the complete year but unverified, or are there missing months?

Data processing and statistical analyses

- Why have you have used the term lethality rate as opposed to case fatality rate?

Results

Table 2:

- In the title, please clarify if this refers to Imported Malaria Cases in Cabo Verde.

- I recommend restructuring this table to focus on the characteristics of imported cases, and separating the data for incidence, mortality, and lethality rates.

- Correct the spelling of "sex" (not "sexe").

Figure 2:

Is this a form of stacked chart? Please specify in the title or figure caption if it is.

Figure 4:

The percentages and their corresponding colours difficult to follow. Consider using a continuous colour scale, and perhaps include a supplementary table with the exact percentages.

Figure 5:

Countries are listed on the x-axis, but for Guinea, the capital, Conakry, is also included – please standardise this.

Discussion

Third Paragraph:

- Please define all acronyms, such as ECOWAS.

- In the sentence, “In addition to the ‘brother countries,’ neighbouring countries such as Senegal, a member of ECOWAS, and Guinea-Bissau and Cape Verde” – it suggests that Cape Verde is a neighbouring country to itself, which is confusing. Please clarify.

- The last sentence of the third paragraph is difficult to understand. Please improve clarity.

Ninth Paragraph:

- Please add references when discussing the interval between symptoms and visiting healthcare.

Tenth Paragraph:

- Please define all acronyms, such as KAP.

Second to Last Paragraph:

- Use a lowercase "p" for Prediction.

- "So." should not be a sentence on its own.

- "Now, rather than ever" should be changed to "Now more than ever."

Reviewer #2: First, thank you to the authors for this valuable and timely manuscript. The paper is well-organized, rich in surveillance data, and offers important insights into post-elimination malaria risks in Cabo Verde. This topic is highly relevant for other recently malaria-free countries grappling with the challenge of imported cases.

That said, a major revision is recommended to sharpen the focus, improve clarity, and enhance policy utility. Please see detailed comments below.

Major Comments

1. Overly descriptive, needs more analytical depth

The manuscript tends to summarise data without offering strong interpretation or synthesis.

For example, trends in figures and tables are restated in the text without drawing broader conclusions or identifying actionable implications.

Suggested Fix: Strengthen the discussion of what the observed trends mean for malaria reintroduction risk and preparedness strategy.

2. Too lengthy and repetitive in sveral sections

The introduction and discussion include repetitive paragraphs, particularly around WHO certification, climate change, and the need for sustained vigilance.

Suggested Fix: Reduce length by eliminating repetitive phrases and focusing on unique contributions of the study.

3. Missing or underdeveloped aspects

Delay in care-seeking behavior (average of 4.2 days for imported cases) is mentioned, but not analysed in-depth. Why is this delay occurring, and what are the implications?

The gender and age profile (82% male, >90% over 25) is noted but not discussed—these trends could inform targeted interventions for high-risk groups.

There is little discussion on P. ovale or mixed infections, though small in number, these may impact treatment approaches.

4. Link between imported cases and local reintroduction needs more emphasis

The paper acknowledges that 16 introduced cases were reported in 2024 but does not analyse how they may be linked to specific importation events.

Suggested Fix: Consider adding a case example or flow pathway illustrating how an imported case might lead to onward transmission.

5. Figures and tables need condensation

Several figures (especially those showing month-wise or origin-wise data) could be combined or moved to supplementary materials.

Simplify visuals and keep the focus on Figures 1 (trend), 3 (forecast), and 4 (geographic clustering).

Minor Comments

Title: Consider simplifying to: “Imported Malaria in Cabo Verde (2010–2024): Risks to Post-Elimination Stability”.

Abstract: Rephrase lines 34–35 – forecast ranges (-23.1 to 84) suggest low model certainty. Consider framing more cautiously.

Line 146–158 (Study area): This could be shortened or partially moved to supplementary material.

Line 300+: The discussion about climate change is helpful but currently too generic. Trim it and cite relevant local studies or specific Cabo Verde projections if available.

Check small inconsistencies: e.g., Angola mentioned as both contributing 18.6% and 19.0% (Lines 37 and 550).

Reviewer #3: SUMMARY

The authors have conducted a study on characterising imported cases of malaria in Cabo Verde and presented a comprehensive discussion on the challenges and priorities for preventing the disease re-establishment in the country. The methods are clearly described and appropriate to the analysis. The results are well presented, with the data and analyses supporting all results. The results are contextualised in relevant literature with a good discussion on the implications of the findings. I commend the authors on an interesting and important piece of work.

While the manuscript is generally well written and easy to follow, it would benefit from a proofread and revision of a few sentences that are difficult to interpret. I have highlighted some of these examples below.

PROOF-READING

Line 63 - Should “Indigenous” be capitalised?

Line 77 - “Outages” seems like an odd choice of words

Line 117 - I suggest removing “the” so the sentence reads more easily: “This will support decision-makers…”

Line 129 - This should be a comma to denote a thousand rather than a period (I see you have used commas lower down so presumably this is a typo). Also please check for consistency of using either a comma or space to denote thousands and a period to denote decimal points (e.g. in Table 2 you are using a comma to denote decimal points.)

Line 177 - I suggest using the term “Case fatality rate” instead of “Lethality rate” (also in Table 2) as it is a more commonly used term. Also revise this in Table 2.

Line 207 - Full stop missing at the end of the sentence

Table 2 - English spelling: Cases by sex (not sexe) and Mix or Mixed (not Mixte)

Line 3 on page 14 - check whether you’ve introduced the acronym KAP yet.

Paragraph 3 on page 15 - No need to put ‘climate change’ in inverted commas.

General: Please do a read through of the document to check for unnecessary use of capitalisation in the middle of sentences.

General: Please check for consistency of using either lowercase or uppercase when referring to figures in the text.

RESULTS

Suggest adding a footnote to Table 1 with the definitions of the 4 key terms (imported, local, introduced, relapsed).

It seems that Table 2 is only providing details on Imported cases, not all cases. This should be made clear in the heading.

When commenting on the species of imported cases: Which species are detected on the tests and do all tests used have the ability to detect different species?

“Over the years, the number…” - perhaps rephrase to “Over the remaining years…” for clarity.

On page 10 when discussing Figure 1, do you have any thoughts on why 2018 was such a high year? It appears to be an anomaly and I was wondering whether there are potential explanations. This is just something I was curious about when reading, but does not need to be addressed if there is no clear comment to add to it.

When presenting Figure 3, the predictions have a wide confidence interval. I think this should be raised in the limitations or addressed somewhere in the manuscript. Also, could the authors clarify what the interpretation of values below zero is for the CI when the unit is cases.

In Figure 5, remove the decimal point in the Year legend.

In the results text above Figure 5: “Fewer countries contributed…” - It seems like there are 24 other countries identified as contributing to imported cases (i.e. quite a lot, not fewer). Please reword this for accuracy/clarity.

DISCUSSION

Overall, the authors have presented a comprehensive discussion section raising important considerations, challenges, and priorities for maintaining malaria-free status. I think it would strengthen the discussion section to include a brief summary of the key finding at the beginning of the discussion; and to include a comment on limitations of the study.

While the discussion section is generally well written and easy to follow, there are a few sentences that need revision. I have shown these below - please could the authors rewrite to improve clarity and check grammar.

Pg 12 paragraph 1: “In addition to the “brother countries”, neighbouring countries such as Senegal, a member of ECOWAS, and Guinea-Bissau and Cape Verde, which have an open border with the archipelago, is a destination of choice.”

Pg 12 paragraph 1: “It is known that livelihood activities and population mobility influence the epidemiology of vector-borne disease and, which may increase the risk of malaria cases are barriers to and influence malaria risk in elimination settings [30,31].

Pg 12 paragraph 2: Suggest minor changes to improve readability (using ‘most’ three times in the sentence makes it hard to digest). Change: “Similarly to elsewhere on the African continent, most of the imported malaria cases originated from some of the most African malaria-endemic countries, where P. falciparum is the most predominant parasitic species in humans [38,39].” To: “Similarly to elsewhere on the African continent, the majority of the imported malaria cases originated from some of the most malaria-endemic African countries, where P. falciparum is the predominant parasitic species in humans [38,39].

Pg 12 paragraph 3: In: “1- the “importation of cases” of malaria, from boats that stop by the endemic malaria countries before achieving the archipelago and docking in its ports, especially in São Vicente.” I suggest changing ‘achieving’ to ‘reaching’ or ‘arriving at’.

Pg 15 paragraph 3: “With rising temperatures and competent vectors present, there is a risk of their infestation by infected travellers besides the widespread insecticide resistance.”

Pg 15 paragraph 3: “So. Now, rather than ever, Cabo Verde should reinforce active imported cases surveillance and not ‘relax’ due to the low absolute number of cases.”

Reviewer #4: This manuscript presents a description of surveillance data and malaria importation over a 15 year period in Cabo Verde, during which malaria elimination was certified. The paper is a useful commentary on the challenges of maintaining local elimination and will be useful for other countries with similar goals.

Main comments

- Methods, line 142-144. What was the method used to determine the location an infection was acquired? I would expect that the definition would include a requirement for travel within a defined timeframe prior to the diagnosis.

- Methods. I note that the terms introduced case and relapse case are used in the results, but definitions of these terms are not provided in the methods.

- Methods, data processing. It would be informative to have some further details on data collection and processing beyond the single sentence on line 175. Presumably case data are not recorded at health facilities in Excel? Alternatively, are there other publications on the surveillance system in Cabo Verde which could be cited here?

- I am not convinced that the joinpoint regression adds much to the manuscript. The methods and results are not presented in any detail, and this does not seem to add any further information over what could be interpreted in a descriptive analysis. I recommend either presenting these in more detail or removing.

- Lines 190-103. Similarly, the time series forecasting is not presented in sufficient detail for the method to be replicable. Please either provide further details (e.g. how seasonality and “randomness” were accounted for in the model), the R packages used, any covariates, the unit of analysis (annual national data?) and any other assumptions or data handling procedures to develop and validate the model approach.

- Line 199. I believe that the “spatial model” mentioned in line 199 is more accurately reported as a descriptive analysis, unless there are any further spatial statistical methods used.

Additional minor edits

- Abstract. Please ensure that the acronym APC is explained. The abstract should be interpretable as a stand-alone piece of information.

- Abstract. “The most common countries of infection were…” could perhaps be better phrased as “the most common sources of imported infection were…” since a country is not itself infected.

- Introduction, lines 106-108. It may be helpful in this mention of different data collection and surveillance approaches to highlight that the country’s level of transmission is also relevant – the system in a moderate burden country is likely to be very different to that of a country close to elimination

- Introduction, line 112. “Over the years all cases registered in the country have been the subject of an epidemiological investigation”. It would be helpful to state the time period here. Is this a recent development or has it been going on for decades?

- Methods, line 176. Please confirm the source of your population denominator data (national census?)

- Methods, line 177. “Case fatality rate” is a more commonly used terminology than “lethality rate”

- Line 206. If the data are publicly available as stated here, please identify the source so that other researchers may also have access to these data.

- Line 223. Should this sentence refer to introduced cases rather than imported cases?

- Results. Is a “semester” a transmission season, a 6-month period, or something else?

Reviewer #5: The manuscript presents an interesting area of study and offers valuable lessons for malaria-eliminating countries. However, it's unclear how:

• the authors have defined the importation of cases mainly from outside the country, which is OK, but the analysis is retrospective. It would have been helpful to see an analysis of importation both between the islands within Cape Verde and from outside the country. As it stands, the manuscript primarily suggests that imported cases are from outside the country, which is OK and aligns with WHO guidelines for preventing re-establishment following certification. However, since this is a retrospective analysis, it should address both in-country and external sources of importation. This will help identify the most vulnerable entry points or islands.

• This is not described in the MS, Could the authors provide more detailed categories in their importation framework, such as the location of residence and the most likely location of infection?

• Additionally, the authors should check for spelling errors throughout the manuscript.

**Do you want your identity to be public for this peer review?** For information about this choice, including consent withdrawal, please see our Privacy Policy

Reviewer #1: No

Reviewer #2: **Yes: ** Harsh Rajvanshi

Reviewer #3: No

Reviewer #4: No

Reviewer #5: No

---

## [Decision Letter · Decision Letter 1]

9 May 2025

PGPH-D-25-00516R1

Imported malaria in Cabo Verde (2010-2024): Risks to Post-Elimination Stability

Dear Dr. DePina,

Thank you for submitting your manuscript to PLOS Global Public Health. After careful consideration, we feel that it has merit but does not fully meet PLOS Global Public Health’s publication criteria as it currently stands. Therefore, we invite you to submit a revised version of the manuscript that addresses the points raised during the review process.

As the reviewers point out, the difference between a malaria case and a confirmed malaria case is not clear and there is a need for further proof-reading. In addition, to facilitate reproducibility, it would be good to include details of the model selection/comparison (e.g. how seasonal effects were chosen for the Holt-Winters prediction model) and how goodness-of-fit was assessed. 

We look forward to receiving your revised manuscript.

Kind regards,

Michele Nguyen

Academic Editor

Journal Requirements:

Additional Editor Comments (if provided):

Reviewers' comments:

Reviewer's Responses to Questions

**Comments to the Author**

Reviewer #1: (No Response)

Reviewer #3: (No Response)

Reviewer #4: All comments have been addressed

Reviewer #5: All comments have been addressed

publication criteria?

Reviewer #1: Yes

Reviewer #3: Yes

Reviewer #4: (No Response)

Reviewer #5: Yes

3. Has the statistical analysis been performed appropriately and rigorously?

Reviewer #1: Yes

Reviewer #3: Yes

Reviewer #4: (No Response)

Reviewer #5: Yes

4. Have the authors made all data underlying the findings in their manuscript fully available (please refer to the Data Availability Statement at the start of the manuscript PDF file)?

Reviewer #1: Yes

Reviewer #3: Yes

Reviewer #4: (No Response)

Reviewer #5: Yes

5. Is the manuscript presented in an intelligible fashion and written in standard English?

Reviewer #1: Yes

Reviewer #3: Yes

Reviewer #4: (No Response)

Reviewer #5: Yes

Reviewer #1: Dear Authors,

Thank you for the updates — they certainly enhance the clarity and overall quality of the manuscript.

I have one remaining comment:

I appreciate the inclusion of Table 1; however, I’m still unclear on the distinction between a Malaria case and a malaria case, confirmed. In both instances, it appears that any positive diagnostic test defines the case. Apologies if I’m overlooking something.

Reviewer #3: Thank you to the authors for their comprehensive responses and updates to the manuscript. All feedback has been appropriately incorporated, except the addition of study limitations in the discussion. On line 488, this section is introduced but does not cover any limitations.

While the manuscript is generally well written, it requires another proofread as there remains some editing to be done (examples given below). Once this is complete, I recommend the paper for publication.

PROOF-READING EDITS

General: check for consistency in use of malaria vs Malaria throughout.

Line 112 - Suggest rephrasing to “Malaria surveillance data…” i.e. remove “It is also crucial to note…” as you make a similar assertion in line 116.

Line 120 - Typos: “decades” and “pre-elimination”

Line 142 - 142: Could you merge these? I.e. just say “...western region of the islands, which is more temperate and less mountainous.”

Line 180 - “It is…” not “It’s”

Line 186 - 187: Suggest rephrasing: “Table 1 provides definitions of key terms used in this study.”

Line 196: Authors use the term “National Malaria Control Program” and introduce the acronym at this point but have also referred to it as “National Malaria Program” earlier (e.g. line 173). Please check for consistency and introduce the acronym in the first instance.

Line 235: Is “anonymised” meant to be removed? If so, also remove the “and” before “publicly”.

Line 257 - 259: Suggest rephrasing for clarity “During the study period, there were 26 recrudescence cases. The majority (18/26) were reported during the 2017 outbreak, with sporadic recrudescence cases also reported in 2013, 2016, 2019 and 2020.”

Line 262: Do you mean “recrudescence” instead of “recurrences

Line 261 - 264: There seems to be a lot of duplication here. Please rewrite or remove duplicated sentences.

Table 2 - Please check for consistency of upper and lower case throughout the table (e.g. sometimes you use title case “Mortality Rate” and sometimes sentence case “Incidence rate”).

Line 282: Suggest removing this sentence or rephrasing to “The remaining months had between x% and 10.4% of the total annual reported cases.”

Line 295: Consider rephrasing to be more concise: “Throughout all years, the majority of cases were reported in Santiago Island (261; 68.1%). Praia, the capital, accounted for 50.4% (209/383) of imported cases nationwide and 80.0% (206/261) of those within Santiago Island (Fig 4).”

Line 326: Indigenous should not be capitalised.

Line 337: Perhaps rephrase to just list the most prominent countries of origin. Eg “ Guinea-Bissau, Angola, Senegal and Nigeria were the most common countries of origin for imported malaria cases.” You address the language connection lower down. Please check this interpretation is correct as there seem to be other countries with more imported cases.

Line 353 - 355: It seems like you’re saying these are risk factors and are also barriers to risk. Please rephrase.

Line 367: Remove “the facility of transport” and just start the sentence with “The increase or flights and mobility…”

Line 381: Remove “the” … i.e. “...between Cabo Verde and endemic malaria countries.”

Lines 382 and 386 - why is “importation of cases” in quotation marks?

Reviewer #4: (No Response)

Reviewer #5: (No Response)

**Do you want your identity to be public for this peer review?** For information about this choice, including consent withdrawal, please see our Privacy Policy

Reviewer #1: No

Reviewer #3: No

Reviewer #4: No

Reviewer #5: No

---

## [Editor Report · Decision Letter 2]

19 May 2025

Imported malaria in Cabo Verde (2010-2024): Risks to Post-Elimination Stability

PGPH-D-25-00516R2

Dear Dr DePina,

We are pleased to inform you that your manuscript 'Imported malaria in Cabo Verde (2010-2024): Risks to Post-Elimination Stability' has been provisionally accepted for publication in PLOS Global Public Health.

Best regards,

Michele Nguyen

Academic Editor